# Natural Farming Practices for Chemical-Free Agriculture: Implications for Crop Yield and Profitability

Ranjit Kumar [ID], Sanjiv Kumar *, BS Yashavanth, Nakeertha Venu [ID], PC Meena, A Dhandapani and Alok Kumar

ICAR-National Academy of Agricultural Research Management, Hyderabad 500030, India
* Correspondence: sanjiv.kumar@naarm.org.in

**Abstract:** The "Green Revolution" (GR) technology-induced agricultural intensification has transformed India from food scarcity to a food surplus country. However, this has also resulted into several adverse repercussions. Increased application of chemical fertilizers and pesticides with stagnating/declining crop productivity has dovetailed with uncertain market conditions and climate change effects which has resulted in un-remunerative agriculture. Consequently, farmers have fallen into the debt trap due to the rising cost of crop production apart from health hazards due to serious exposure to harmful chemical pesticides. Natural Farming (NF), an agro-ecological approach to farming is believed to be an effective way to counter some of these challenges. The present paper presents field-level farmers' experiences of NF adoption in three states of India—Andhra Pradesh, Karnataka, and Maharashtra. The study was conducted during February–March 2019 by surveying 295 NF adopted and 170 non-NF adopted farmers. It was found that NF practice has been followed by some farmers for more than 10 years but others have adopted during the recent past. There is variation in the practice followed by the NF farmers. There are farmers who are using Farm Yard Manure (FYM). A solid form of *jeevamritha* (liquid concoction of microbial inoculants) called as *ghanajeevamritha* was also found to be used by farmers in Andhra Pradesh. It was observed that non-NF yields are superior to NF yield without FYM. In most crops, however, NF with FYM had a greater yield than NF without FYM and non-NF farms. There has been a decrease in the variable cost and a marginal increase in the market price of NF produce. The study suggests that natural farming may be seen as one of the alternative practices which has potential to rejuvenate the agro-ecosystem, besides cost saving for the individual farmers.

**Keywords:** natural farming; zero budget natural farming; green revolution; agroecology; low-input; chemical-free farming



## 1. Introduction

In the history of mankind, there have been many advancements emerging from a belief followed by the discovery of scientific evidence. Our ancestors developed methods to survive on the earth through different raw materials and tools. There was no understanding of science, and everything was generated by the trial-and-error method. This was the same in agriculture; if we look back to our roots, the farming system was completely dependent on on-farm inputs. However, with the timeline of evolution, the exponential growth of the population demanded speedy growth in agricultural production that triggered the emergence of Green Revolution technologies (GRTs) introducing high yielding varieties of crops which were responsive to a higher dosage of chemical fertilizers and irrigation and motivated the farmers to go for intensive monocropping. Food production in India has increased 5–10 times since the introduction of GRTs in the mid 1960s [1]. Despite its success, the input-intensive 'Green Revolution' has concealed substantial externalities in recent decades, harming natural resources, human health, and agriculture itself [2].

Evidence also suggests that a higher dosage of chemical fertilizers has resulted in pest resurgence. The presence of pesticide residues multi-times higher than the prescribed

limit in contaminated drinking water and/or air, biodiversity losses, nitrate leaching and pollution of groundwater, and heavy metal accumulation in the soil are quite common in intensive agriculture regions [3]. Easy availability of pesticides has led to suicidal death in rural India, while indiscriminate application has resulted in the decline of pollinators [4]. To reduce the burden of the rising cost of crop cultivation, successive governments (state and central) have increased input subsidies [5,6] for the farmers in various forms, such as the fertilizer subsidy (Appendix A, Figure A1), free electricity for irrigation, interest subvention on agricultural credit, and a premium subsidy for crop insurance. For instance, during 2016–2019, agricultural consumers were allotted 75% of total electricity subsidies in India. In 2019 alone, the direct subsidy amounted to INR 1103.91 billion and a cross-subsidy of at least INR 750.27 billion [7]. Many state governments have provided free electricity to the farmers, while others have given 75–80% subsidies for agricultural purposes. Institutional credit to agricultural sectors in 2020–2021 was INR 15,754 billion. An interest subvention of even 3% would indicate a subsidy of INR 473 billion [8]. For crop insurance, central government and state governments contributed towards a premium payment worth INR 570 billion in 2021 [9]. All the above issues appear to form a perfect storm for the small-holders and tenant farmers as well as for the long-term sustainability of Indian agriculture. As a result, Indian farmers are increasingly trapped in a never-ending cycle of debt adding to further distress.

Natural Farming (NF) is a one-of-a-kind chemical-free farming approach that is regarded as an agroecological approach [10]. Agro-ecological practice is believed to have been initiated by a Japanese farmer, Masanobu Fukuoka, the local customized version of which has been introduced in India by one of the Indian agriculturists Sh. Subash Palekar in the mid-1990s in the name of 'Zero Budget Natural Farming (ZBNF)'. The core of Natural Farming practices is the application of *Jeevamritha* and *Beejamritha*. *Jeevamritha* is a liquid fermented concoction of cow dung, cow urine, jaggery, pulses flour, and bund soil mixed with water, which contains a large number of beneficial microbes that act as a bio-stimulant promoting the activity of soil microorganisms as well as phyllospheric microorganisms when applied to the field/foliage. *Beejamritha* is also *Jeevamritha* without water that is used for seed treatment. Beneficial microbes are expected to colonize the roots and leaves of germinating seeds, assisting in the healthy growth of the plants. Other important components include *Achhadana* (bio-mulching), intercropping, and use of local seeds. Furthermore, natural farming also promotes various home-made formulations (*neemastra*, *agniastra*, and *bramhastra*, among others) that act as bio-pesticides [11]. These are used to control pests such as mealy bugs, sucking pests, fruit, stem and pod borer, leaf roller, etc. NF has been found to partially improve soil health and this may be because of quick building of heterotrophic microbial communities and flora and the increase in soil organic matter [12–15]. Some studies have indicated a decrease in yield [14,16,17] whereas others showed no decrease [18,19].

In the last couple of years, the government of India has promoted natural farming in big way to promote chemical-free farming. The Prime Minister of India in his address to the nation on the 76th Independence Day of India stated 'ZBNF is a promising tool to minimize the dependence of farmers on purchased inputs, it reduces the cost of agriculture by relying on traditional field-based technologies which also leads to improved soil health' [19]. Schemes such as—*National Mission on Natural Farming, Paramparagat Krishi Vikas Yojana* (Conventional Agriculture Development Scheme) under the sub-mission of *Bharatiya Prakritik Krishi Paddhati (BPKP)*, Andhra Pradesh Community Natural Farming (APCNF), *Mission Organic Value Chain Development for North Eastern Regions (MOVCDNER)*, etc., are popularizing the adoption of natural farming among the farmers in different parts of the country. Under the BPKP scheme, a provision of financial assistance of INR 12,200/ha (Approx. 147 USD/ha) for 3 years is made for cluster formation, capacity building and continuous handholding, certification and residue analysis. The Indian Council of Agricultural Research (ICAR), the apex research body, has initiated a study on the evaluation of NF on certain crops [20,21]. The popularity of NF has drawn the attention of many sections

in society. It is estimated that more than 500,000 hectares of land in India across different states are currently being cultivated under natural farming [22] and it is expected that this may expand to bring 14 million hectares of land under natural farming by 2025 under the PKVY scheme [23]. The scaling up of NF may not only depend on the farming practices, but social factors such as social movements, public policies, markets, pedagogical processes, leadership, and discourse also play a key role [24–26]. Farmer-focused and farmer-led knowledge exchange is a key driver of the sustained spread of NF practice [27].

Keeping the above description in the background, this study attempts to answer core questions such as: In what form is natural farming being practiced by Indian farmers? What would happen to the crop yield and income of the farmers? How are the natural farming practices helping the agro-ecology? The present study is based on field survey conducted in three major states of India (Karnataka, Andhra Pradesh, and Maharashtra) where the adoption of natural farming is believed to be high. The study has examined the adoption pattern of different components of NF by the farmers, and estimated the crop yield and farm income under NF practices as compared to existing farming practices.

## 2. Materials and Methods

### 2.1. Outline of the Research

Being one of the very few studies in this area, the research team faced several challenges in identifying and selecting the farmers who could be considered Natural Farming adopters (NF-adopters). Similarly, natural farming practice is not a uniform and standardized practice; therefore, the adoption of package of practices also varies significantly, making a comparative study quite challenging.

The field survey was conducted during February–May 2019 in three states of India—Andhra Pradesh, Karnataka, and Maharashtra. Based on consultation with agricultural universities in respective states, districts were identified where the NF practice has been adopted by considerable number of farmers. The farmers were selected using snowball sampling in the sample districts. Those farmers who used *jeevamritha* and did not use any chemical fertilizer and/or pesticide in last one year were considered NF-adopters, while others were non-adopters (non-NF). In each state, two or three districts with a higher proportion of farmers taking up natural farming practices were selected (Table 1 and Figure 1). Non-NF farmers were also selected from the same villages for comparison purposes.

Information from the sample farmers was elicited through personal interviews. A pre-tested and structured survey schedule was used to survey 120 NF-adopted and 60 non-adopted farmers in Andhra Pradesh and Maharashtra, respectively. In Karnataka, it was extremely difficult to find NF-adopted farmers during the field survey. The majority of villages had only one or two NF-adopted farmers. As a result, a comprehensive survey covering 29 villages was conducted to identify 55 NF adopted farmers. Non-NF farmers were also chosen from the same villages for comparison purposes.

**Table 1.** Distribution of sample farmers in the study.

| State | District | No. of Villages Covered | NF-Adopted Farmers | Non-NF Adopted Farmers | Total Sample Farmers |
|---|---|---|---|---|---|
| Andhra Pradesh | Vishakhapatnam | 5 | 60 | 30 | 90 |
| | Vizianagaram | 5 | 60 | 30 | 9 |
| Karnataka | Mandya | 10 | 32 | 24 | 56 |
| | Ramanagara | 8 | 7 | 10 | 17 |
| | Tumakuru | 11 | 16 | 16 | 32 |
| Maharashtra | Parbhani | 6 | 60 | 30 | 90 |
| | Hingoli | 7 | 60 | 30 | 90 |
| Total sample size | | 52 | 295 | 170 | 465 |

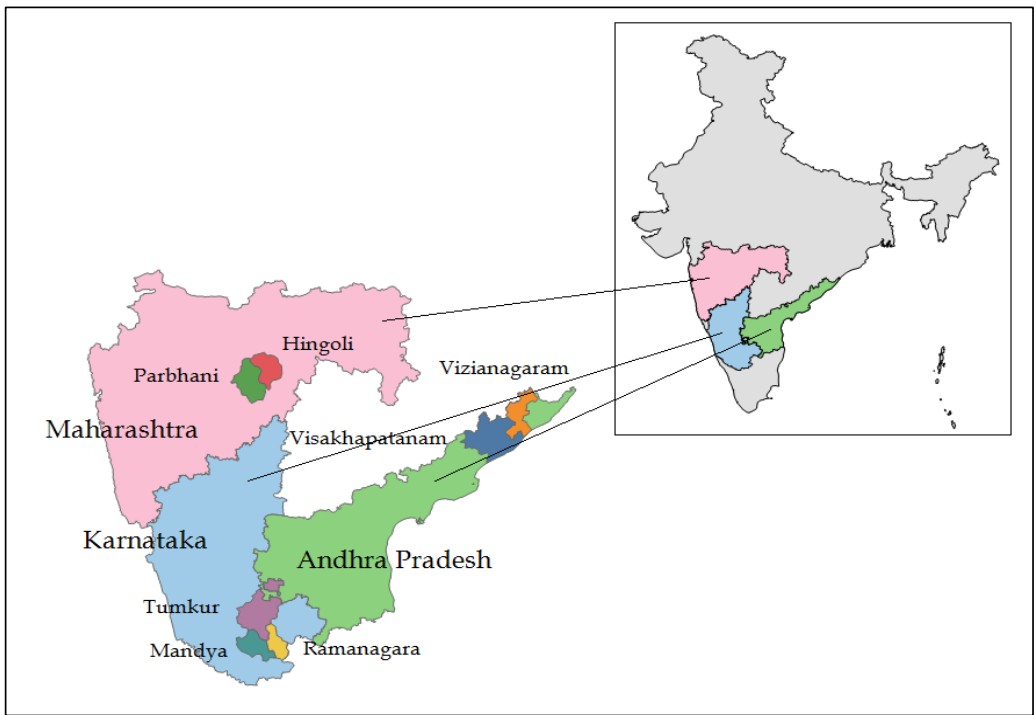

**Figure 1.** States and sample districts for the study.

The collected data from the sample farmers were analyzed using descriptive statistics. One way ANOVA was used to test the difference in yield for selected crops under NF and non-NF.

Benefit:Cost Ratio (B:C Ratio) was used for selected crops under NF.

$$\text{B:C Ratio} = \frac{\text{Yield (q)} \times \text{Market Price (₹/q)}}{\text{Total Cost (₹/q)}}$$

q-quintal (1 q = 100 kg)

Change in economic parameters as well as change in B:C ratio in NF from Non-NF for respective crops was also estimated.

$$\Delta_{\text{EP}} = \frac{\text{EP}_{\text{nf}} \times 100}{\text{EP}_{\text{nnf}}}$$

where

$$\Delta_{\text{EP}} - \text{Change in economic papameter}$$
$$\text{EP}_{\text{nf}} - \text{Economic parameter value under NF}$$
$$\text{EP}_{\text{nnf}} - \text{Economic parameter value under Non} - \text{NF}$$

### 2.2. Study Area Description

The study area comprising of two–three districts each of the three states had the following geographical characteristics (Table 2):

**Table 2.** Geographical characteristics of the study area.

| Particulars | Andhra Pradesh | Karnataka | Maharashtra |
|---|---|---|---|
| Districts under study | Vishakhapatnam, Vizianagaram | Mandya, Ramanagara, Tumakuru | Parbhani, Hingoli |
| Annual rainfall (mm) | 1100–1200 mm | 580–720 | 945–960 |
| Main irrigation source | Tank and canal | Borewell and canal | Canal |
| Soil type | Red clay, sandy loam, clay laom, loamy, coastal sandy | Black, red, sandy, and sandy loam soil | Deep black, shallow soil |
| Major crops | Paddy, sugarcane, black gram, green gram, groundnut, finger millet, mango, vegetables | Paddy, sugarcane, horse gram, cowpea, groundnut, finger millet, mango, vegetables | Soybean, cotton, sorghum, pigeon pea, green gram, black gram, chickpea, vegetables |

*2.3. Demographic Characteristics*

The farmers in the study area included both young and middle-aged farmers. The majority of the farmers were in their forties (>30 years) and had at least a decade of farming experience, whether they practiced Natural Farming (NF) or non-NF (conventional/chemical farming). In Andhra Pradesh, the proportion of young farmers (30 years old) practicing NF was higher than the proportion of non-NF farmers. Farmers practicing NF outnumbered non-NF farmers in Karnataka between the ages of 30 and 50. However, the proportion of young farmers who practiced NF was negligible. The majority of NF farmers in Maharashtra were between the ages of 40 and 50. When the educational qualifications of the NF farmers in all three states were examined, the majority of them had at least intermediate education or equivalent. However, in Karnataka, a disproportionate number of NF farmers were graduates or higher. Illiterates took a major share among non-NF farmers compared to NF farmers in all the three states. The average family size of the sampled farmers in all three states was discovered to be between 4 and 6, with 2–3 members engaged in farming in each family.

**3. Results and Discussion**

*3.1. Results*

3.1.1. Adoption of Natural Farming

Natural farming has been practiced by farmers in some parts of the country for several decades, though it has recently gained popularity. Among the sampled farmers, 27% of NF farmers in Karnataka had practiced NF for more than ten years. In Maharashtra and in Andhra Pradesh, the majority of NF farmers (66% and 85%) had 3–6 years and less than 3 years of experience, respectively (Figure 2). The government of Andhra Pradesh established Rythu Sadhikara Samstha (RySS), a not-for-profit company in 2014. RySS implements community managed natural farming in the state which has resulted in the adoption of NF by a large number of farmers in the state in recent years.

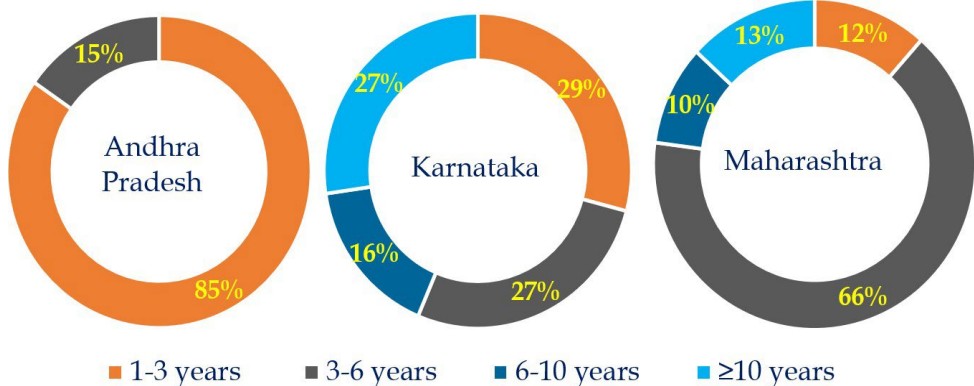

**Figure 2.** Experience of practicing natural farming by the sample farmers.

Intercropping is a major recommended practice in Natural Farming as it reduces soil stress by reducing the mining of only specific nutrients from the soil, as in the case of a solo/mono crop. In some cases, intercrops/mixed crops complement each other in terms of nutrient cycling. Nonetheless, despite its recommendation, only 26%, 45%, and 17% of NF farmers in Andhra Pradesh, Karnataka, and Maharashtra, respectively, practice inter-/mixed crops. (Figure 3).

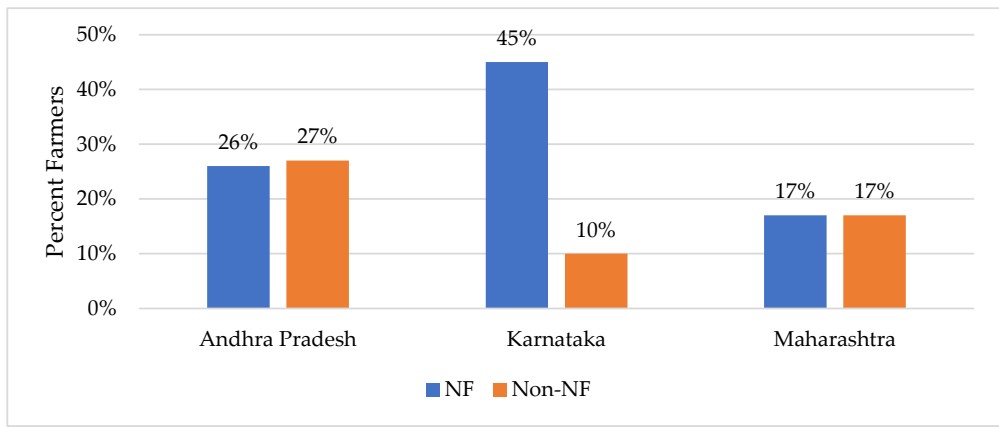

**Figure 3.** Percentage of sample farmers following mixed cropping or intercropping in natural farming.

The low percentage of inter-/mixed cropping is due to the fact that paddy is the major crop in the study area and is preferably cultivated as a single crop. Karnataka (45%) had the highest rate of inter/mixed cropping among the study states. It was observed that almost the same proportion of farmers in Andhra Pradesh and Maharashtra practiced inter/mixed cropping in both NF and non-NF. However, in Karnataka, only 10% of non-NF farmers practiced inter/mixed cropping.

In Andhra Pradesh, a solid form of *jeevamritha* called as *ghanajeevamritha* was used by the farmers. The farmers applied *ghanajeevamritha* (0.5–1 t ha$^{-1}$) by broadcasting before sowing in the field. *Beejamritha* for seed treatment was used depending on the crop as well. There was no fixed quantity of *jeevamritha* used in the field. Farmers in Karnataka were found to use higher quantities of *jeevamritha* than other state farmers. It was as high as 3000 L per acre in Paddy. It was also found that farmers who used Farm Yard Manure (FYM) in the field applied lower quantities of *jeevamritha*. Although, NF practice does not advocate the use of FYM, it was found that NF farmers used FYM in the field before sowing. In Andhra Pradesh, 52% of farmers used FYM in all crops, while 36% did not use it at all. The remaining 12% of farmers used FYM in specific crops such as sugarcane and paddy. More than 52% of NF farmers in Maharashtra were found to not use FYM in their fields. FYM was used in 20% of farmers' fields. Sugarcane is a high-value crop, and more than 80% of sugarcane-growing NF farmers in Andhra Pradesh and Karnataka used FYM in their fields. In Maharashtra, 63% of turmeric growing NF farmers used FYM in their turmeric fields (Figure 4).

Mulching, an important component of NF, was found to be followed by farmers depending on the crop and the availability of mulching material. Farmers in Andhra Pradesh used azolla for paddy mulching, which was not observed in Karnataka. Farmers used a variety of mulching materials, including live mulch crops such as cow pea, other farm waste, straw, and sugarcane/coconut trash.

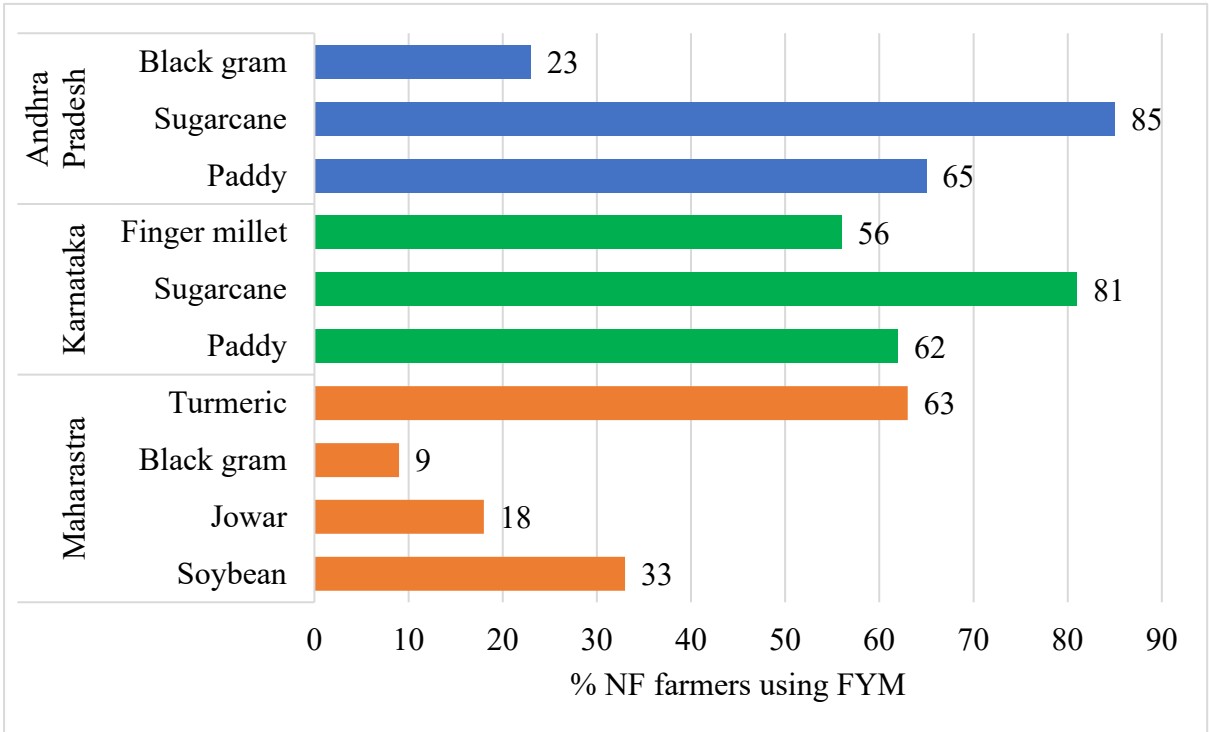

**Figure 4.** Crop-wise application of Farm Yard manure (FYM) by NF adopters.

3.1.2. Yield Variation

Crops such as paddy, sugarcane, cotton, soybean, blackgram, sesamum, finger millet and several vegetable crops were found to be cultivated in the study area. Perennials such as coconut, arecanut, mango, cashew, and banana were also cultivated in some parts. The yield of major crops was worked out for the three farming practices viz. NF with FYM (1–2 t ha$^{-1}$ in paddy and 2–4 t ha$^{-1}$ in sugarcane at every/alternative year(s)), NF without FYM, and non-NF. On comparison, it was observed that non-NF yields were superior to NF yield without FYM. In most crops, however, NF with FYM had a greater yield than NF without FYM and non-NF farms. It can be deduced from the preceding discussion that natural farming practices alone could not yield as much as conventional farming, but supplemented with a small amount of FYM, crop yields were invariably higher than those from conventional/chemical farming, providing a clear picture of farmer yield sustainability. To compare the yield of crops under non-natural farming (non-NF), natural farming (NF) with FYM and natural farming without FYM, a one-way ANOVA was used (Appendix A—Tables A1 and A2).

In case of black gram in Andhra Pradesh, NF without FYM had a significantly lower yield than NF with FYM (At $p < 0.1$). Moreover, in the case of paddy in Karnataka, NF without FYM had a significantly lower yield than non-NF as well as NF with FYM (At $p < 0.05$). The difference in yield could not be established in other cases due to the absence of a critical number of farmers adopting different practices. During the field survey, crop yield over the previous three years with the NF farmers was also explored. It was carried out to see if the yield of important crops grown under NF in the past had improved or not. Almost all of the crop yields were found to be more or less consistent in all three states over the three years 2016–2018 (Figure 5).

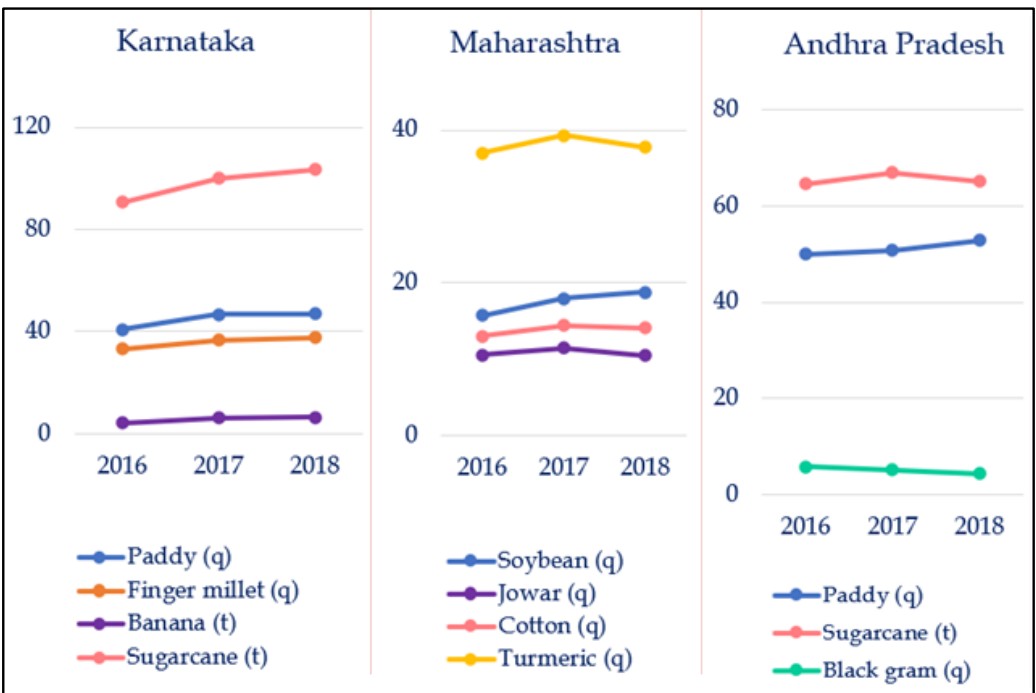

**Figure 5.** Trend in yield (t/ha or q/ha) of major crops under NF from 2016 to 2018 {t-tonne (1 t = 1000 kg), q-quintal (1 q = 100 kg)}.

3.1.3. Benefit–Cost Analysis of Natural Farming

The study examined the use of various inputs in the cultivation of major crops and estimated the paid-out cost and return for NF and non-NF farms. Table 3 details the various costs incurred in the cultivation of major crops in the selected states. The percentage of the corresponding cost with respect to non-NF crops is also presented. Material cost includes costs incurred in seed, *jeevamritha*, *Beejamritha*, FYM, and a pest controlling solution for NF farmers, whereas for non-NF farmers, it is mainly seed, fertilizer, FYM, and pesticide. Operational cost includes cost of land preparation and labor including harvesting. These two were added to arrive at the total paid-out cost in both the cases.

In Andhra Pradesh, the material cost used by NF farmers in case of paddy and sugarcane was about 85 and 96%, respectively as those for non-NF farmers. Although it was less than that of non-NF, it was higher than NF farms in other states. This may have been because large number of farmers apply purchased FYM and *ghanajeevamritha* in their field, as only 40 per cent of NF farmers have indigenous cows, and they depend on purchased materials.

The operational cost in the case of the same crops was closer to the non-NF counterparts. Hence, the total variable cost was lowered by only 5% in paddy and 8% in sugarcane. In case of black gram, the NF farmers could reduce the total variable cost by around 55%. This could be due to the reduction in material cost as only 23% farmers apply FYM as compared to paddy and sugarcane in which 65% and 85% farmers apply FYM (Figure 6). Farmers are able to obtain a marginally higher price for NF produce than non-NF produce. Except sugarcane, the B:C ratio was found to be improved in Andhra Pradesh for NF farmers.

In Karnataka, NF farmers have mostly home-made *Jeevamritha* and *Beejamritha* which has resulted in a drastic reduction in the material cost to around 24% in paddy, 45% in sugarcane, and 26% in finger millet. The operational cost is a little less than their non-NF counterparts. Farmers here could be able to obtain a maximum of 150% more price as in case of paddy and minimum 50% more as in case on finger millet. It should be noted that NF farmers are mostly cultivating *Rajamudi*, *Rathnachudi*, and *Bangara Sanna* which have a

high market price. Here, the Benefit:Cost ratio (B:C ratio) has increased by 3–4 times than that of non-NF.

**Table 3.** Benefit–Cost comparison for major crops in selected states.

| | Andhra Pradesh | | | | | |
|---|---|---|---|---|---|---|
| | **Paddy** | | **Sugarcane** | | **Black Gram** | |
| **Particulars** | **NF** | **As % of Non-NF** | **NF** | **As % of Non-NF** | **NF** | **As % of Non-NF** |
| No. of sample farmers | 118 | 59 | 35 | 6 | 22 | 6 |
| Material costs (INR/ha) | 9050 | 84.82 | 26,780 | 95.53 | 856.62 | 39.10 |
| Operational costs (INR/ha) | 25,960 | 98.51 | 39,473 | 89.44 | 6525 | 58.46 |
| Total variables cost (INR/ha) | 35,011 | 94.56 | 66,253 | 91.81 | 7382 | 55.28 |
| Yield (q/ha) | 53 | 104.2 | 65 | 88.63 | 4.5 | 81.82 |
| Market price (INR/q) | 1525 | 112 | 2480 | 99.2 | 3765 | 104.58 |
| B:C ratio | 2.3 | 123.4 | 2.43 | 95.79 | 2.29 | 154.44 |
| | Karnataka | | | | | |
| | Paddy | | Sugarcane | | Finger millet | |
| Particulars | NF | As % of Non-NF | NF | As % of Non-NF | NF | As % of Non-NF |
| No. of sample farmers | 42 | 22 | 18 | 14 | 15 | 23 |
| Material costs (INR/ha) | 4031 | 23.72 | 11,638 | 45.53 | 2314 | 25.73 |
| Operational costs (INR/ha) | 17,491 | 91.66 | 28,914 | 92.36 | 17,688 | 97.48 |
| Total variables cost (INR/ha) | 21,522 | 59.66 | 40,552 | 71.31 | 20,002 | 73.71 |
| Yield (q/ha) | 47 | 83.65 | 103 | 103.48 | 38 | 134.9 |
| Market price (INR/q) | 3945 | 264.51 | 5200 | 198.7 | 3700 | 153.14 |
| B:C ratio | 8.6 | 370.69 | 13.2 | 270.7 | 6.97 | 279.91 |

| | Maharashtra | | | | | | | | | |
|---|---|---|---|---|---|---|---|---|---|---|
| | Soybean | | Jowar | | Cotton | | Turmeric | | Chickpea | |
| Particulars | NF | As % of Non- NF | NF | As % of Non- NF | NF | As % of Non-NF | NF | As % of Non-NF | NF | As % of Non-NF |
| No. of sample farmers | 61 | 46 | 69 | 33 | 37 | 34 | 57 | 21 | 52 | 23 |
| Material costs (INR/ha) | 6838 | 65.6 | 3869 | 55.4 | 6595 | 37.8 | 45,121 | 68.5 | 4905 | 69.6 |
| Operational costs (INR/ha) | 12,851 | 105 | 9593 | 102.8 | 19,934 | 115 | 28,468 | 92 | 8241 | 81.2 |
| Total variables cost (INR/ha) | 19,689 | 86.9 | 13,462 | 82.5 | 26,529 | 76.2 | 73,589 | 76 | 13146 | 76.4 |
| Yield (q/ha) | 19 | 103.6 | 10.5 | 100.8 | 15 | 88.3 | 38 | 93.8 | 15 | 84.9 |
| Market price (INR/q) | 3208 | 103.7 | 3091 | 115.1 | 5021 | 101.2 | 5957 | 92.8 | 4576 | 109.8 |
| B:C ratio | 3.13 | 123.7 | 2.42 | 140.67 | 2.84 | 117.24 | 3.04 | 114.72 | 4.3 | 122.15 |

q-quintal (1 q = 100 kg).

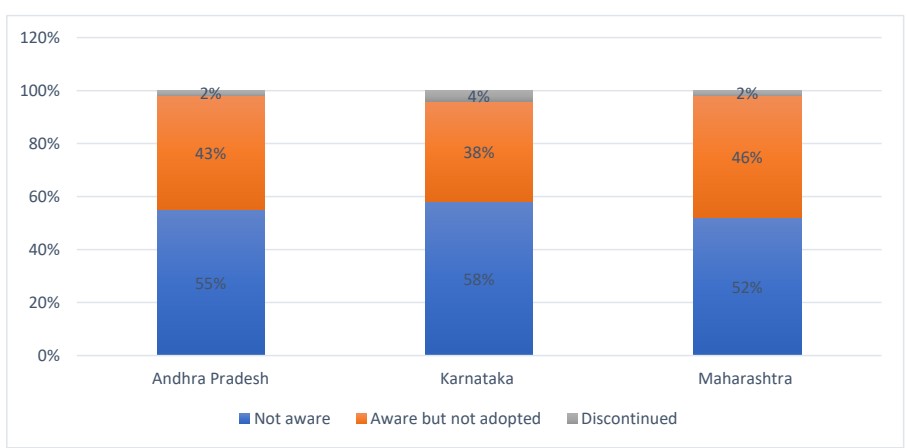

**Figure 6.** Awareness about NF among non-NF farmers (in % of non-NF farmers surveyed).

In Maharashtra, there was a decrease in variable cost for all the major crops which was reduced by around 13% in soybean to 24% in cotton, turmeric, and chickpea. There was a marginal increase in market price for all the crops as farmers cannot access the niche market for the sale of the NF produce. The B:C ratio was also improved by more than 15% in all the crops.

Although the study could not measure the exact saving of water from natural farming practices due to the variety of crops being grown by the adopter farmers. However, all the adopter farmers expressed that they applied 0–2 irrigations depending upon the crops grown through NF practices, as compared to 3–4 irrigations under non-NF farming. This has two important implications—one, in terms of saving irrigation water which is becoming one of the scarcest natural resources for agriculture, and second, it also saves electricity for which many state governments give subsidy in terms of free electricity for irrigation or highly subsidized electricity charges for irrigation purposes.

3.1.4. Benefits Perceived by NF Farmers

Farmers perceived many benefits of NF. Majority of the farmers in Andhra Pradesh, (81%) and Maharashtra (60%) believed that the yield of crops increased (Table 4). In Karnataka, 22% felt that the yield has increased whereas 56% felt that it decreased and 20% felt that it remained the same.

**Table 4.** Benefits perceived by Natural Farming farmers.

| Perceived Benefits | Percent Farmers | | |
| --- | --- | --- | --- |
| | Andhra Pradesh | Karnataka | Maharashtra |
| Crop yield | | | |
| High | 81 | 22 | 60 |
| Same | 17 | 20 | 16 |
| Lower | 2 | 56 | 24 |
| Cost of cultivation | | | |
| High | 1 | 7 | 9 |
| Low | 86 | 93 | 91 |
| Taste of produce | | | |
| Better | 91 | 89 | 89 |
| Same | 9 | 11 | 11 |
| Selling price | | | |
| High | 22 | 96 | 81 |
| Same | 69 | 4 | 19 |
| Lower | 1 | 0 | 0 |
| Sometimes high/low | 8 | 0 | 0 |

NF practice reduces the cost of cultivation which was felt by 86% of farmers in Andhra Pradesh and more than 90% in Karnataka and Maharashtra. As far as produce quality and taste are concerned, around 90% in all the selected states found that NF produce had a better quality than non-NF produce. In Andhra Pradesh, farmers were not able to access the designated market for the sale of NF produce; hence, the produce was sold in the same market at almost same price. In Karnataka and Maharashtra, farmers were able to access the designated market where the produce can fetch a higher price.

### 3.1.5. Awareness among Non-NF Farmers

Though farmers perceived several benefits out of NF practice, more than 50% farmers among non-NF category in three selected states were not yet aware of NF. However, only 2–4% farmers discontinued and reverted to the conventional system of farming owing to no obvious benefits of NF (Figure 6). Lower crop yield and no immediate control over pests and diseases were found to be the reasons for discontinuation. Decreased landholding and no proper support from family members were also the reasons for discontinuation in a few cases.

### 3.1.6. Reasons for Non-Adoption among Non-NF Farmers

In Andhra Pradesh, non-availability of inputs due to very low percentage of ownership of indigenous cows was one of the major reasons for not adopting NF (Figure 7).

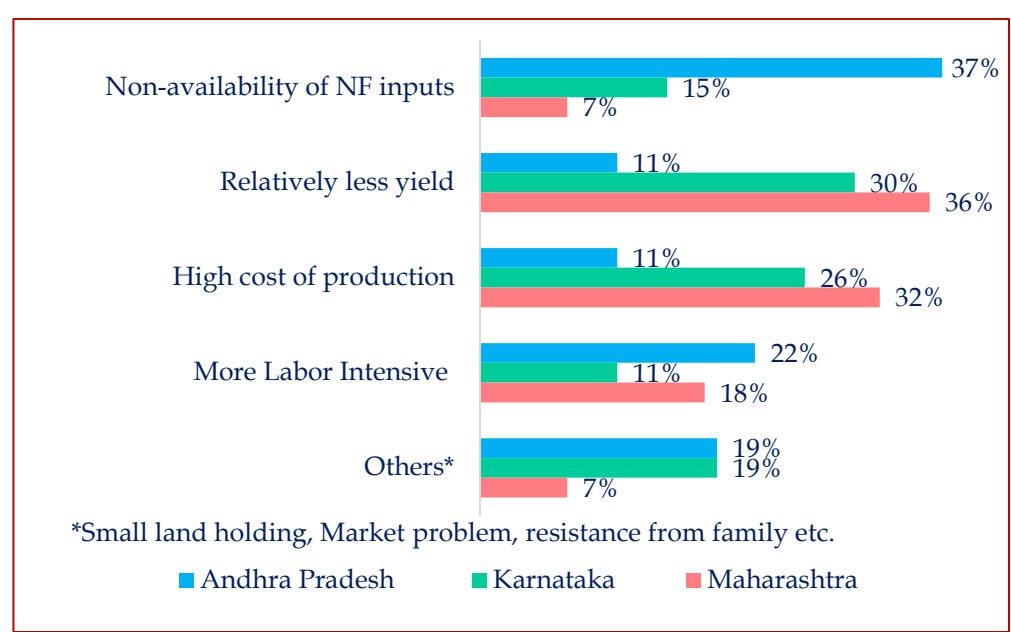

**Figure 7.** Reasons of non-NF farmers for not adopting NF (in % of surveyed non-NF farmers).

However, 60% of NF farmers also do not have indigenous cows, rather they buy all inputs from the village's nutrient shops, while a few collect them from fellow farmers. The expectation of poor crop yield was also one of the reasons for the non-adoption of NF by non-adopted farmers (more than 30% in Karnataka and Maharashtra). NF was perceived to be more labor intensive and regular monitoring is required from the part of farmers. Preparation of *jeevamritha*, *Beejamritha*, as well as farm operations require regular attention by the farmer. It also discourages farmers from adopting NF. The farmers also expect higher prices for the NF produce considering it is free from chemicals. Hence, the non-availability of designated market for NF produce (as in the case of organic produce) has driven reluctance towards NF adoption.

### 3.2. Discussion

The number of farmers adopting NF has increased, which could be due to the initiatives of government of India as well as some from the state government. The government of Andhra Pradesh promoting Andhra Pradesh Community Natural Farming (APCNF) has led to an increase in substantial NF farmers in the recent past which is very much supported by the study result. The variation found in NF practice indicates that farmers in different areas have modified the recommended practice depending on the availability of a particular input as well as the results experienced by them by using a particular input. For instance, farmers in Andhra Pradesh use a solid form of *jeevamritha* called *ghanajeevamritha* which was not found in any other state. In fact, farmers were found to use FYM which is not considered as part of NF practice. It was found that the NF alone does not give a higher yield. However, when FYM is added, the yield increases. There are studies which show that the yield decreases when NF is followed [14,16–18]. Some other studies have indicated an increase in yield [19]. The increased use of pesticides has affected the health of farmers and has pushed them into debt trap [28,29]. However, as NF does not require chemical inputs such as fertilizers and pesticides, the input cost is reduced as compared to non-NF. This eventually results in a better net profit if the yield is not affected much.

As input costs fall, farmers are less likely to take out loans, which frees them from the debt cycle and saves the government money by drastically reducing the fertilizer subsidies. There was a slow transition in the conversion to natural farming despite of natural farming schemes and missions. Moreover, as farmers perceive the produce as better in taste and quality with a chemical-free quality, it may result in a better price realization. However, lack of market access is hindering price realization [14]. Some other factors such as the non-availability of input and the greater labor-intensiveness may be deterrent factors towards NF adoption.

### 3.3. Limitations

As the study was based on a field survey of farmers, it had its own limitations inherent in social science studies. Additionally, as there was no official list of NF farmers, sample selection was heavily dependent on those who declared themselves as following the practice. The information collected from farmers was based on recall basis, which may sometimes have had recall error.

## 4. Conclusions and Recommendations

### 4.1. Conclusions

Natural Farming was found to be widespread in Andhra Pradesh with the majority joining the bandwagon during the last 5 years, whereas in Karnataka and Maharashtra, although the adoption of NF though started more than 15 years back, it was very much sporadic. The crop yield in NF was not higher as compared to conventional farming. However, when supplemented with FYM/*Ghanajeevamritha*, the crop yield improved significantly. It was also evident that there was a substantial reduction in input cost in NF as compared to non-NF due to non-use of expensive agro-chemicals. This has resulted in a significant reduction in the cost of cultivation of all the crops. This has resulted in better profitability (B:C ratio) for NF farmers.

The benefits as perceived by NF farmers are manifold, ranging from a lower cost of cultivation, better quality and taste to a premium price. Though the premium price benefit is not experienced by many farmers, it creates a new market opportunity for tapping a middle-class customer segment, who aspire to consume chemical-free produce, but are hesitant to pay exorbitant prices for organic produce.

### 4.2. Recommendations

Farmers have been successfully using NF in various forms in some regions for a long time. Intuitively, it echoes the possibility of regenerating nutrients required for plant growth under NF by activating various microorganisms and adding biomass to the soil.

Natural Farming may not be looked at as yield-enhancing farming practices, but as one of the alternative practices particularly for those regions which are rainfed and have less intensive farming practices. It also helps in increasing the farmers' income through cost reduction by saving market-based farm inputs. The NF produce may be recognized as niche produce free from chemicals and with a better quality and taste. It will help the farmers in realizing higher price for the produce.

The practice seems to be good for human health as well as environmental health. However, systematic research is required to validate the long-term sustainability of the production system to examine the nutrient availability in the soil and to the crops. Hence, there is a requirement for generating scientific evidence before scaling out in different agro-climatic regions with different crop combinations in order to prove this hypothesis and theory of change.

**Author Contributions:** Conceptualization, R.K., S.K., B.Y. and P.M.; Formal analysis, S.K., A.D. and A.K.; Investigation, S.K., B.Y. and P.M.; Methodology, R.K. and B.Y.; Project administration, R.K.; Supervision, R.K.; Visualization, S.K.; Writing—original draft, R.K., S.K., B.Y. and N.V.; Writing—review and editing, R.K., S.K., N.V., A.D. and A.K. All authors have read and agreed to the published version of the manuscript.

**Funding:** The research study was funded by NITI Aayog (National Institution for Transforming India), apex public policy think tank of the Government of India. Sanction No. O-15012/33/18-Research.

**Institutional Review Board Statement:** The study was approved by institute research council with assigned project code as AGED/NAARM/CCL/2019/004/00158.

**Informed Consent Statement:** The project report and the draft paper has been submitted to the PME cell of the institute.

**Data Availability Statement:** Available on request from the corresponding author.

**Acknowledgments:** The authors thank NITI Aayog, Government of India for sponsoring the study. We also place on record the support and guidance of the Director, ICAR-NAARM. The project also got support of other researchers as well as engaged several field investigators and young professionals to conduct the field survey. We acknowledge their support.

**Conflicts of Interest:** The authors declare no conflict of interest.

## Appendix A

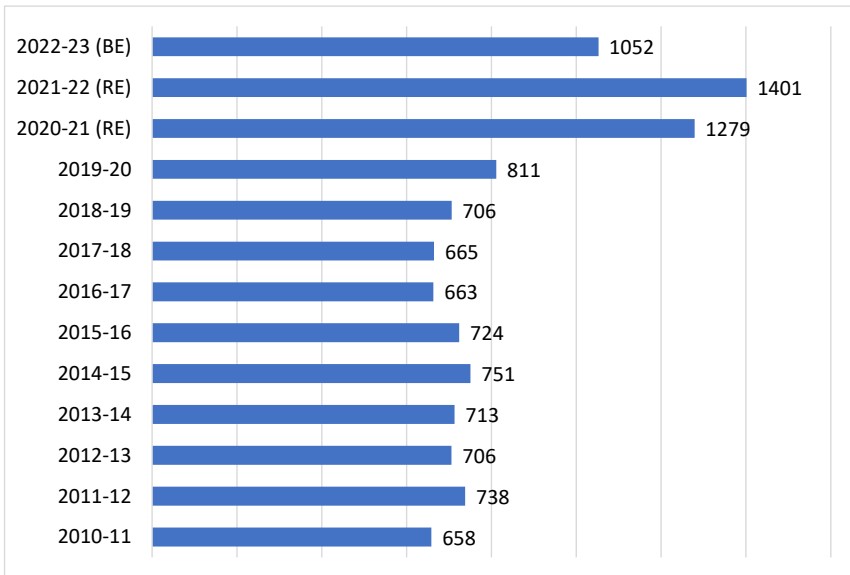

**Figure A1.** Total Subsidy on all Fertilizers from 2010–2011 to 2021–2022 (Billion INR). (BE—Budget Estimate, RE—Revised Estimate).

**Table A1.** ANOVA result of yield comparison for black gram in Andhra Pradesh.

| | | | | | | | | |
|---|---|---|---|---|---|---|---|---|
| **Yield (q/ha)** | | | | | | | | |
| | *n* | **Mean** | **Std. Deviation** | **Std. Error** | **95% Confidence Interval for Mean** | | **Minimum** | **Maximum** |
| | | | | | **Lower Bound** | **Upper Bound** | | |
| Non-NF | 11 | 5.40 | 3.96 | 1.19 | 2.74 | 8.06 | 1.25 | 11.67 |
| NF without FYM | 26 | 3.77 | 2.51 | 0.49 | 2.75 | 4.78 | 0.83 | 10.00 |
| NF with FYM | 8 | 6.40 | 3.18 | 1.12 | 3.75 | 9.06 | 0.37 | 10.00 |
| Total | 45 | 4.64 | 3.15 | 0.47 | 3.69 | 5.58 | 0.37 | 11.67 |

| ANOVA | | | | | |
|---|---|---|---|---|---|
| | Sum of Squares | df | Mean Square | F | Sig. |
| Between Groups | 50.95 | 2 | 25.48 | 2.778 | 0.074 |
| Within Groups | 385.23 | 42 | 9.17 | | |
| Total | 436.18 | 44 | | | |

| Post Hoc Tests (Tukey HSD) | | | | Dependent Variable: Yield (q/ha) | | |
|---|---|---|---|---|---|---|
| (I) Group | (J) Group | Mean Difference (I–J) | Std. Error | Sig. | 95% Confidence Interval | |
| | | | | | Lower Bound | Upper Bound |
| Non-NF | NF without FYM | 1.63 | 1.09 | 0.301 | −1.01 | 4.28 |
| | NF with FYM | −1.00 | 1.41 | 0.759 | −4.42 | 2.42 |
| NF without FYM | Non-NF | −1.63 | 1.09 | 0.301 | −4.28 | 1.01 |
| | NF with FYM | −2.63 | 1.22 | 0.092 | −5.61 | 0.34 |
| NF with FYM | Non-NF | 1.00 | 1.41 | 0.759 | −2.42 | 4.42 |
| | NF without FYM | 2.63 | 1.22 | 0.092 | −0.34 | 5.61 |

There was a statistically significant difference between groups (At $p < 0.1$) as determined by one-way ANOVA ($F(2.42) = 2.778$, $p = 0.074$). A Tukey post hoc test revealed that NF without FYM (M = 3.77, S.D. = 2.51, $p = 0.092$) had a significantly lower yield than NF with FYM (M = 6.4, S.D. = 3.18). There was no statistically significant difference between non-NF and NF without FYM as well as non-NF and NF with FYM.

**Table A2.** ANOVA result of yield comparison for paddy in Karnataka.

| | | | | | | | | |
|---|---|---|---|---|---|---|---|---|
| **Yield (q/ha)** | | | | | | | | |
| | *n* | **Mean** | **Std. Deviation** | **Std. Error** | **95% Confidence Interval for Mean** | | **Minimum** | **Maximum** |
| | | | | | **Lower Bound** | **Upper Bound** | | |
| Non-NF | 22 | 56.08 | 11.84 | 2.52 | 50.83 | 61.33 | 37.50 | 80.00 |
| NF without FYM | 16 | 38.78 | 9.38 | 2.35 | 33.78 | 43.78 | 20.00 | 50.00 |
| NF with FYM | 26 | 51.92 | 15.66 | 3.07 | 45.60 | 58.25 | 20.00 | 75.00 |
| Total | 64 | 50.07 | 14.54 | 1.82 | 46.43 | 53.70 | 20.00 | 80.00 |

| ANOVA | | | | | |
|---|---|---|---|---|---|
| | Sum of Squares | df | Mean Square | F | Sig. |
| Between Groups | 2924.66 | 2 | 1462.33 | 8.584 | 0.001 |
| Within Groups | 10,391.74 | 61 | 170.36 | | |
| Total | 13,316.40 | 63 | | | |

| Post Hoc Tests (Tukey HSD) | | | | Dependent Variable: Yield(q/ha) | | |
|---|---|---|---|---|---|---|
| (I) Group | (J) Group | Mean Difference (I–J) | Std. Error | Sig. | 95% Confidence Interval | |
| | | | | | Lower Bound | Upper Bound |
| Non-NF | NF without FYM | 17.30 * | 4.29 | 0.000 | 7.00 | 27.61 |
| | NF with FYM | 4.16 | 3.78 | 0.518 | −4.93 | 13.24 |
| NF without FYM | Non-NF | −17.30 * | 4.29 | 0.000 | −27.61 | −7.00 |
| | NF with FYM | −13.15 * | 4.15 | 0.007 | −23.11 | −3.18 |
| NF with FYM | Non-NF | −4.16 | 3.78 | 0.518 | −13.24 | 4.93 |
| | NF without FYM | 13.15 * | 4.15 | 0.007 | 3.18 | 23.11 |

There was a statistically significant difference between groups (At $p < 0.05$) as determined by one-way ANOVA ($F(2.61) = 8.584$, $p = 0.001$). A Tukey post hoc test revealed that NF without FYM (M = 38.78, S.D. = 9.38) had a significantly lower yield than non-NF (M = 56.08, S.D. = 11.84, $p < 0.000$) as well as NF with FYM (M = 51.92, S.D. = 15.66, $p = 0.007$). There was no statistically significant difference between non-NF and NF with FYM.

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
