# Peer review of "Natural Farming Practices for Chemical-Free Agriculture: Implications for Crop Yield and Profitability"

_agriculture, doi:10.3390/agriculture13030647_

Round 1

Reviewer 1 Report

In my opinion, the analysis is at an early stage, representing only an annual analysis and cannot capture the variation generated by climatic conditions.

The ZBNF concept is studied by several researchers, with scientific articles published in prestigious journals, see Ashlesha Khadse, Peter M. Rosset and others. I recommend the authors to improve their manuscript by consulting a larger number of publications, relevant in the field.

The three questions formulated by the authors at the beginning of the manuscript (L83-85) can be transformed into hypotheses and the article created in such a way as to demonstrate whether the hypotheses are validated or not.

The methodology is not clearly presented. The authors present only the demographic aspects of the respondents, but, in the case of the farm analysis, I believe that a presentation of their agricultural specifics is necessary. It must be presented how the farmers were identified and how the questionnaire was applied.

It is necessary to describe the three study areas from the point of view of climatic conditions, soil types, predominant crops.

The article can be improved by:

introducing a new chapter, DISCUSSIONS

formulation of study limitations

formulating recommendations, based on the results obtained.

Reviewer 2 Report

Major recommendations

1.     The Introduction Section. According to Instructions for Authors, in the Introduction Section, the current state of the research field should be carefully reviewed and key publications cited. However, there are only 13 citations. The analysis is rater weak. It must be expanded. The authors should review worldwide experience. It is desirable for the authors to strengthen their arguments and deepen the analysis of previous studies.

2.     Scientific gaps should be revealed.

3.     The authors should underline the novelty of this study.

Minor recommendations

1.     I suggest the authors improving the Abstract Section. The abstract need the briefly description of main methods and the results.

2.     Lines 52-53: “…. free electricity for irrigation1, interest subvention on agricultural credit2, and premium subsidy for crop insurance3.” All information must be presented in the main text. The text should be without footnotes.

3.     Line 76: “… a provision of financial assistance of ₹12,200/ha for 3 years is made” I would recommend to present currency rating of national Indian currency to USD. It will be more clear to international readers.

4.     Line 91: ‘2. Methodology ’ The authors should use the recommended style.

5.     The Methodology section has the only subsection. The authors should either combined all parts or add other subsections.

6.     Line 101: “chosen (Table.1 and Fig. 1). “ Table 1 and Figure 1. It is according to the Instructions for Authors.

7.     Line 188: ‘…. with FYM (At p<.1).’ What does it mean? .1? Is it 0.1? Please clarify.

8.     Line 189: ‘… with FYM (At p < .05).’ What does it mean? .05? Is it 0.05? Please clarify.

9.     Line 195: ‘Fig. 5’ All figures and tables should be cited in the main text as Figure 1, Table 1, etc.

10. Line 198: ‘Figure 5. Trend in yield of major crops under NF from 2016 to 2018.’ What are the units for measuring yield?

11. Lines 243-245: Why are they empty?

12. Table 2: ‘B:C ratio ’ What does the B:C ratio stand for? Please explain ‘q/ha’. How many kilograms (ton, etc.) is it? (kg/ha. t/ha).

13. Table 3: Same? May be average? Please clarify.

14. Tables: Tables should be done according to instructions for authors.

15. The Conclusions Section: The prospects for further research should be more specifically and clearly described.

Reviewer 3 Report

The manuscript with title "Natural farming practices for chemical-free agriculture: implication cations on crop yield and profitability" is writing about the field level farmers’ experiences of natural farming adoption in three states of India. The topic is interesting for agriculture researchers. The following questions may improve the quality of the manuscript.

1.       Line 40-41 Please give some reference.

2.       Why you choose those farmers who used jeevmritha and did not use any chemical fertilizer and/or pesticide in last one year? Why not the last two or three years?

3.       Please revised the Fig. 3 for better expression.

4.       Please added some discussion of the results in the section of “Results”.

5.       Please added your survey question as the supplement file.

6.       The authors did not sufficiently identify the limitations of their research. Please clarified.

Round 2

Reviewer 1 Report

The authors have considerably improved the material, taking into account the previous recommendations.

Congratulations!

Author Response

Thank you for your critical comments in earlier review.

Reviewer 3 Report

this manuscript had been improved 

Author Response

(The authors gave the same response as above.)
